# Effect of Number of Layers on Tensile and Flexural Behavior of Cementitious Composites Reinforced with a New Sisal Fabric

Adilson Brito de Arruda Filho [1,2], Paulo Roberto Lopes Lima [1,3,4], Ricardo Fernandes Carvalho [1], Otavio da Fonseca Martins Gomes [5,6] and Romildo Dias Toledo Filho [4,*]

1. Postgraduate Program in Civil Engineering, Federal University of Bahia, Salvador 40170-115, BA, Brazil; adilson.arruda@ufrb.edu.br (A.B.d.A.F.); prllima@uefs.br (P.R.L.L.); ricardoc@ufba.br (R.F.C.)
2. Center for Exact and Technological Sciences, Federal University of Recôncavo da Bahia, Cruz das Almas 44380-000, BA, Brazil
3. Postgraduate Program in Civil and Environmental Engineering, State University of Feira de Santana, Feira de Santana 44036-900, BA, Brazil
4. Postgraduate Program in Civil Engineering, Federal University of Rio de Janeiro, Rio de Janeiro 21945-970, RJ, Brazil
5. Centre for Mineral Technology, Rio de Janeiro 21941-908, RJ, Brazil; ogomes@gmail.com
6. Postgraduate Program in Geosciences, National Museum, Federal University of Rio de Janeiro, Rio de Janeiro 20940-040, RJ, Brazil
* Correspondence: toledo@coc.ufrj.br

**Abstract:** The use of fabric in reinforcing cement-based materials expands their applications for various types of construction elements. Additionally, employing renewable sources of plant-based fabrics contributes to reducing the environmental impact of the construction industry. However, the variability in the properties of plant fibers and fabrics necessitates prior studies to confirm their effectiveness as reinforcement materials. In this study, a new sisal fabric was produced and utilized as reinforcement in cement-based matrix composites. The sisal fibers, yarns, and fabrics produced were tested under direct tension. Five composites were manufactured by manual lamination, with reinforcement ranging from one to five layers, and were subjected to direct tension and flexural testing. The results indicate that, while the fiber shows brittle failure, the yarn and fabric exhibit a gradual loss of strength after reaching the maximum tension. All composites display strain-hardening and deflection-hardening behavior, with multiple cracking and an increase in tension and deformation before rupture. The mechanical properties exhibited improvement with an increase in the number of layers, and composites with four and five layers displayed distinct behavior, demonstrating increased stiffness after the occurrence of multiple cracking and a better mechanical performance, qualifying them for use as a construction element.

**Keywords:** sisal fabric; strain-hardening behavior; natural fiber; textile reinforced mortar

## 1. Introduction

The incorporation of fabrics into cementitious matrices has revolutionized the production of construction elements, offering new mechanical properties and diverse shapes. This advancement has expanded the applicability of these materials, fostering more efficient construction practices. From early use in reinforced mortar with metal meshes to the current era of 3D fabrics, cementitious textile composites have continually evolved in terms of performance and durability.

The application of these composites spans from creating roofing and façade elements for buildings [1] to structurally reinforcing beams [2] and historical masonry restoration [3]. However, the pressing need for sustainable solutions, aiming to reduce the consumption of non-renewable raw materials, energy, and waste during production, has driven the exploration of plant-based fabrics as alternatives for composite reinforcement [4]. Various plant-based fabrics, including hemp [5], jute [6,7], flax [8], and sisal [9], have been employed

to enhance the flexural strength and toughness of cementitious composites. After cracking, the tension is transferred to the fabric, which has an adequate anchorage length to redistribute stresses without being pulled out of the matrix. Consequently, a process of multiple cracking occurs with increased tension, resulting in enhanced rupture deformation.

In addition to the influence of fiber type, the configuration of the fabric and the method of composite production can have a significant impact on fabric–matrix adhesion and, consequently, on mechanical behavior [10]. Claramunt [8] observed that flax fabrics with higher weight and thickness hindered the proper infiltration of cement particles into the porous structure of nonwoven fabrics. Olivito [9] utilized commercial sisal fabrics to reinforce composites, revealing a substantial decrease in tension under direct tension after the initiation of each fissure. These sisal fabrics, designed for use in textiles and carpets, employ very thick threads without spacing between them. Moreover, the fabrics are manufactured with twisted threads, where only a portion of them comes into contact with the matrix along their length. Consequently, there is low fabric–matrix adhesion and internal slippage of the threads not in contact with the matrix.

To address these challenges, this study introduces a novel type of plant-based fabric for cement composites. Finer yarns than their commercial counterparts were produced, featuring straight and aligned fibers to ensure greater fiber–matrix contact along their length. When subjected to tension, the straight fibers quickly mobilize, exhibiting a linear stress–strain behavior, while the twisted yarns display an initial nonlinear phase due to the unrolling process. Additionally, the increased spacing between threads in the new fabric allows the matrix to fill the gaps between them, thereby enhancing adhesion and improving the overall performance of the composite.

As a hypothesis, the addition of a larger quantity of reinforcement usually suggests an enhancement in composite performance. In the case of composites reinforced with plant-based fabrics, where adding more fabric layers affects the production method and the fiber–matrix relationship, studies need to be conducted to confirm the maximum number of layers that can be added without compromising mechanical behavior. Cement-based composites reinforced with one layer [7,9] or three layers of fabric [6], exhibit a significant loss of load after the opening of the first crack and greater crack opening at high deformations, limiting their application. Claramunt [8] observed a slight increase in flexural strength with the increment of the number of layers, respectively, in composites with three and four layers of flax fabrics. The evaluation of composites with a greater number of reinforcement layers was performed by Mawlood [7]. For composites reinforced with four and six layers, a reduction in flexural strength is observed when compared to the reference sample, and a less effective contribution of the reinforcement after the first crack, exhibiting a deflection-softening behavior and a reduction in toughness with an increase in the number of layers. This implies that investigating how multiple fabric layers influence mechanical behavior is essential before using new fabrics as reinforcement for cement-based composites.

Usually, the behavior of cementitious composites reinforced with long fibers or fabrics, where there is sufficient anchorage length for stress transfer, is represented by an initial phase of linear elasticity followed by a process of multiple cracking with a slight increase in stress. The increase in loading results in positive post-yield stiffness, up to a maximum load, followed by negative stiffness and strength degradation until fracture or pull-out of the reinforcement. The prediction of the mechanical behavior of composites under direct tension can be made using linearized models [11], with a good approximation to experimental results until the composites reach the maximum stress. In this work, the trilinear model [12] was used to determine the mechanical behavior of composites with different numbers of layers and to calculate resilience and toughness under direct tension. However, predicting the failure mode of composites and determining post-peak negative stiffness requires the use of more sophisticated methods with theoretical models based on energy [13] or numerical models based on the finite element method [14].

The main objective of this work is to investigate the mechanical behavior of composites reinforced with different layers of an innovative plant-based fabric developed specifically for use with cementitious matrices. The fabric was developed using untwisted sisal yarns with aligned fibers and with sufficient spacing between threads to allow immersion in the cementitious matrix. The mechanical behavior of the fibers, yarns, and fabrics was evaluated through direct tensile testing. Composites reinforced with one to five layers of fabric were produced and tested under direct tension and bending. The behavior under direct tension of composites with different layers was used to validate an analytical model and determine the resilience and toughness of the composites. The failure mode was assessed by monitoring the initiation and propagation of cracks.

## 2. Materials and Methods

### 2.1. Cement Matrix

In order to obtain a durable composite, the matrix of which does not chemically attack the sisal fiber, and which has adequate rheology, the binder constituents used were composed of cement CP V ARI (ASTM Type III) and two mineral additions: 40% of fly ash and 10% of silica fume as partial cement substitutions. Table 1 shows the characteristics of the binder constituents.

**Table 1.** Binder constituents.

| Characteristics | | Cement | Fly Ash | Silica Fume |
|---|---|---|---|---|
| Major chemical component (%) | CaO | 69.77 | 2.06 | 0.17 |
| | $SiO_2$ | 15.89 | 53.33 | 95.3 |
| | $SO_3$ | 4.76 | 1.51 | – |
| | $Al_2O_3$ | 4.35 | 33.23 | 0.04 |
| | $K_2O$ | 1.07 | 3.44 | 1.33 |
| | $Fe_2O_3$ | 3.66 | 4.96 | 0.35 |
| Specific gravity $(g/cm^3)$ | | 3.06 | 2.01 | 2.65 |

The cement matrix in the proportion of 1:1:0.35 (binder:sand:water/binder ratio, by weight) was prepared with the use of river sand with a density of 2.64 $g/cm^3$, a maximum particle size of 1.2 mm, and tap water. A third generation superplasticizer (Glenium 51) with a solid content of 30.9% and a specific gravity of 1.09 $g/cm^3$ was used. The viscosity modifier admixture (VMA) Rheomac UW 410, with a specific gravity of 0.7 $g/cm^3$, at a dosage of 0.05% relative to the binder's weight, was also used in order to avoid segregation during molding. The matrix was designed for a consistency of 400 ± 10 mm in the flow table test without drop, indicating the high flowability of the matrix. The use of mineral additions and additives results in a matrix free from calcium hydroxide [15], ensuring greater durability of the composite while providing sufficient fluidity to impregnate the fabric and ensure adequate fiber–matrix adhesion for structural applications.

The matrix was created using a 20 $dm^3$ capacity, mixed by following the subsequent mixing procedure: Cement and fly ash were initially blended in the mixer, followed by the addition of water and superplasticizer. Silica fume and fine aggregate were introduced and mixed for 4 min at a low speed of 125 RPM. Subsequently, the mixing process was halted for 30 s to eliminate material adhering to the mixer walls. The mixing procedure then resumed for an additional 2 min at a medium speed of 220 RPM. Finally, the VMA was added and mixed for an additional 4 min at 125 RPM.

Table 2 presents the matrix characterization.

**Table 2.** Matrix characterization.

| Property | Water Absorption (%) | Density (g/cm$^3$) | Compressive Strength (MPa) | Splitting Strength (MPa) |
|---|---|---|---|---|
| Value | 4.12 ± 1.10 | 2.09 ± 0.01 | 47.78 ± 0.4 | 3.53 ± 0.20 |

### 2.2. Natural Reinforcement

The sisal fibers (*Agave sisalana*), produced in the state of Bahia, Brazil, were washed in hot water (50 °C) to remove the superficial oils and greases that affect the hydration of the cement, and were brushed to separate the individual fibers.

The yarn used in the weave of the fabric was composed of 10 sisal fibers, arranged straight and aligned. The produced fabric was of a simple woven type, following the model developed by [16]. For its production, a manual loom with a 400 mm width was used. The warp was composed of spaced cotton threads at 5 mm intervals, and the weft consisted of yarns of sisal fibers, containing 10 fibers, spaced 1 mm apart to ensure better impregnation of the matrix in the fabrics (Figure 1).

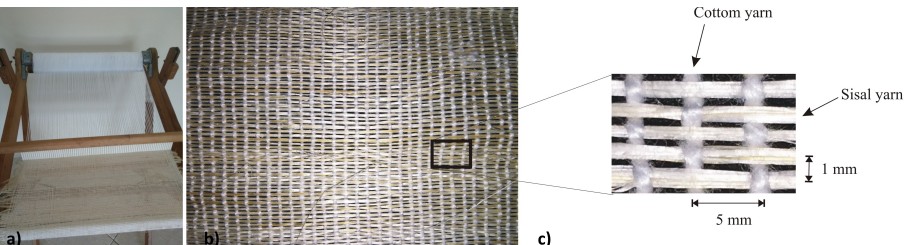

**Figure 1.** Fabric characteristics: (**a**) loom used in production; (**b**) sisal-cotton fabric; (**c**) wire spacing.

To minimize the water absorption of the plant fibers and the subsequent dimensional variation, the fabric underwent a hornification treatment [17]. Initially, the fabric was soaked in water at room temperature (27 °C) for 3 h, allowing it to reach saturation. Following this, the excess water was removed by hanging the fabric on a clothesline for 30 min. Subsequently, the fabric was placed in an 80 °C oven for 16 h to eliminate moisture. To prevent thermal shock, the fabric was allowed to cool in the oven for 4 h and 30 min before removal. This treatment procedure comprised a 24 h cycle of wetting and drying and was repeated 10 times.

### 2.3. Production Method

To create the laminated composites, the matrix was first prepared using a bench-mounted mechanical mixer capable of holding up to 20 dm$^3$. Subsequently, a layer of this matrix was poured onto a metallic mold measuring 400 mm × 400 mm, depicted in Figure 2a. The fabric layer was placed and pressed with a plastic roller until the cement paste emerged on the surface Figure 2b. New layers of matrix and fabric were added until different types of laminates were produced (Figure 2c,d). As shown in Figure 3, the composites were built with one, two, three, four, and five layers of sisal fabric, totaling approximately 10 mm (variation from 9.1 to 11.9 mm).

Figure 3 shows the positioning of the reinforcement layers in the cross-section of the sisal fabric-reinforced cement composites (SFRCC). It is possible to observe the wavy pattern of the fabric, with the sisal yarns interconnected by a wave of cotton threads. Table 3 presents the reinforcement content, by mass, for each laminated composite. Due to the manual process of fabric production, the weight of each fabric was measured before the production of each composite and used in the calculation of the reinforcement content.

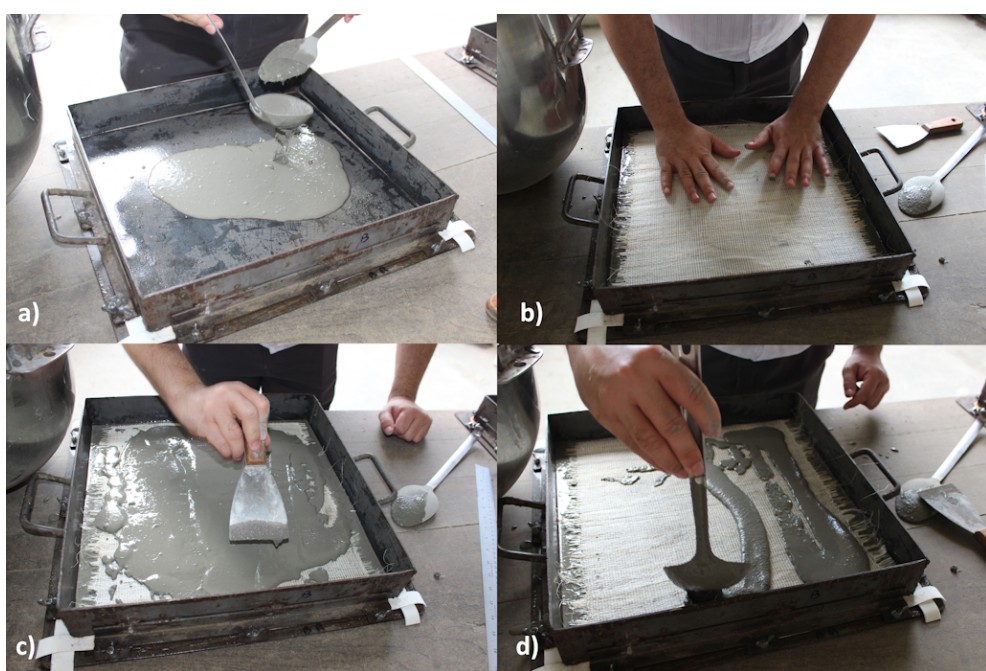

**Figure 2.** Manufacturing the composite reinforced with two layers: (**a**) first layer of matrix; (**b**) fixation of sisal fabric; (**c**) distribution of second layer of matrix; (**d**) second sisal fabric layer and top layer of mortar.

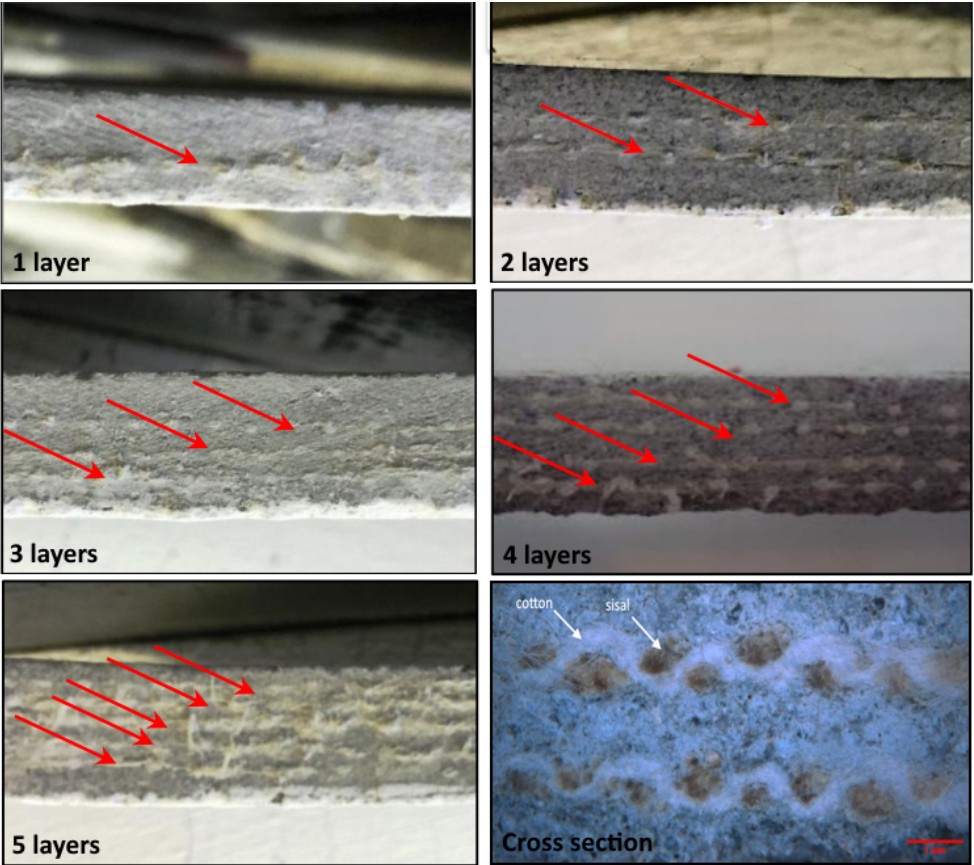

**Figure 3.** Positioning of reinforcement in the cross section of the composite.

**Table 3.** Reinforcement distribution of the composites.

| Group | Sample | Fabric Weight (g/mm$^2$) | Number of Layers | Vf (%) |
|---|---|---|---|---|
| | SFRCC1 | 0.025 | 1 | 2.91 |
| I | SFRCC2 | 0.022 | 2 | 4.11 |
| | SFRCC3 | 0.021 | 3 | 5.48 |
| II | SFRCC4 | 0.024 | 4 | 9.30 |
| | SFRCC5 | 0.024 | 5 | 10.46 |

*2.4. Test Method*

The direct tensile tests of fibers, yarns, and fabric were conducted utilizing a Shimadzu UH-F 100kN universal testing machine in accordance with ASTM C1557. A 1 kN load cell was employed, and the displacement was controlled at 2.0 mm/min during the tests. The cross-sectional area of the reinforcement elements (fibers, yarns, and fabric) was determined by employing their density, which was ascertained through helium pycnometry, in combination with the measured mass of each sample. For the tensile test, the fiber and yarn samples were affixed onto paper molds, and adhesive tape was applied to the specimen's surface at the grips to minimize damage and prevent slippage.

The direct tensile test of the composites was performed on samples with dimensions of 200 mm × 50 mm × 10 mm. Metal plates were glued to the sides of the sample using an epoxy resin-based adhesive to connect them to the machine grips and prevent crushing. The test was conducted on the Shimadzu machine at a speed of 0.5 mm/min. Strains were measured using a 50 mm clipgage placed at the central third of the sample.

Bending tests were conducted using displacement control at a crosshead rate of 0.5 mm/min on the Shimadzu machine with a 1 kN capacity load cell. Three specimens, each with dimensions of 400 × 100 × 10 mm for each mixture, were tested under a four-point bending configuration with a 300 mm span. The flexural strength of the composite was determined from the maximum load carried out by the composite, where the concept of flexural stress ($f_{tf}$) was obtained from the bending formula given by $f_{tf} = 6M/(bd^2)$, where M is the maximum moment of the test specimen and d and b are the depth and width of the specimen's cross section, respectively.

The samples were periodically photographed to observe crack formation during tensile and flexural tests. The fabric–matrix interface was assessed using a binocular microscope.

**3. Results**

*3.1. Tensile Behavior of Reinforcement*

The stress–strain curves obtained in the direct tensile test for the fibers, yarns, and fabrics are shown in Figure 4. It can be observed that the sisal fiber exhibits linear mechanical behavior until reaching the maximum stress, followed by brittle failure, which is characteristic of some vegetable fibers.

The yarn, consisting of a collection of straight sisal fibers, displays linear behavior under stress until the rupture of the initial fiber. This stress–strain behavior differs from the non-linear response observed in yarns created by twisting fibers into a spiral, which undergoes a realignment process of elementary fibers upon initial stress application [18]. Following this point, stress escalation occurs, leading to the progressive rupture of the yarn as the applied force is transferred to the remaining fibers, which also reach their tensile strength. This damage phase has also been corroborated by Gahgah [18] in the direct tensile test of sisal yarn.

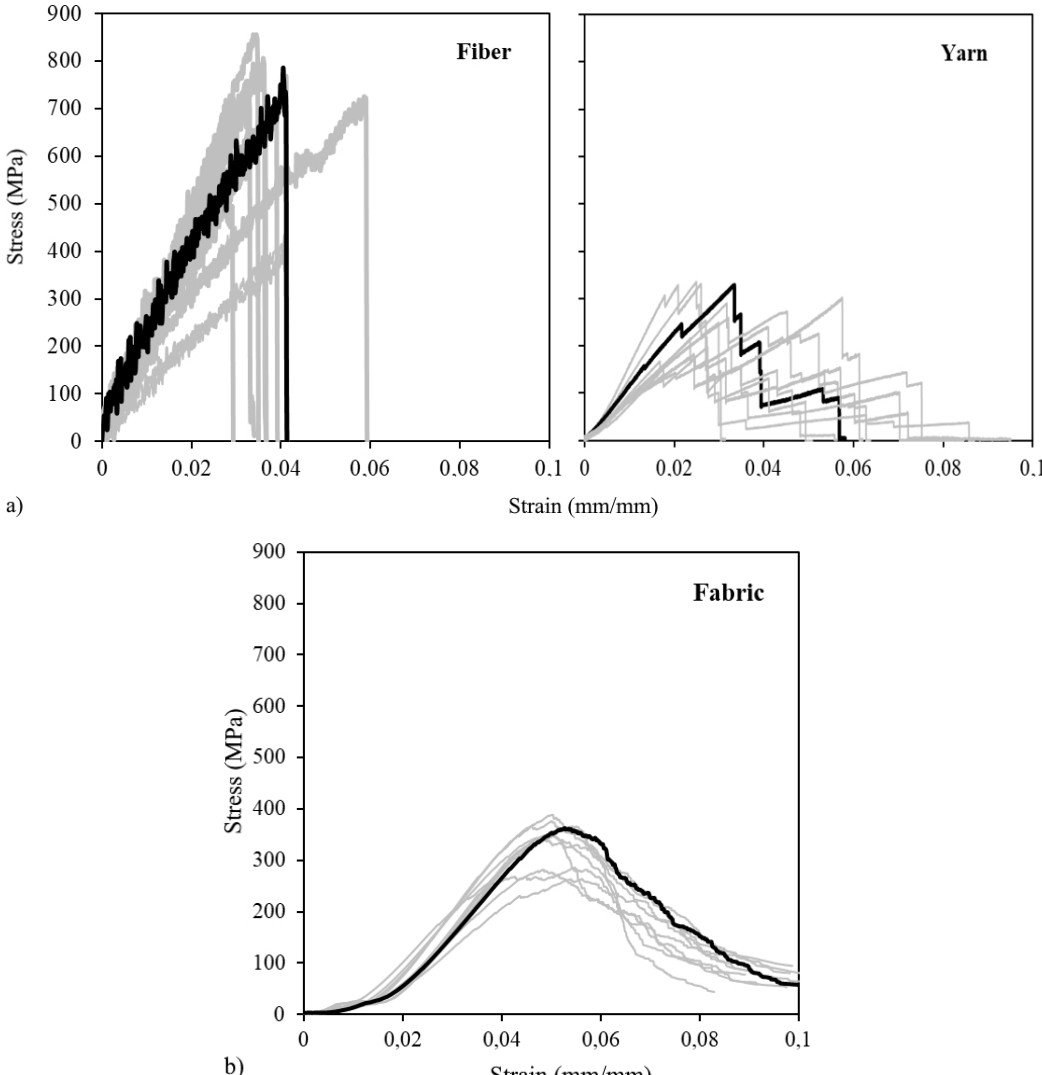

**Figure 4.** Stress–strain of reinforcements: (**a**) fiber and yarn; (**b**) fabric.

The behavior of the fabric is influenced not only by the rupture of the bundle of sisal fibers aligned in the direction of the force but also by the presence of the transversal cotton fiber, forming a waviness in the fabric, as shown in the cross-section of the composite in Figure 3. According to [19], during the tensile strength and elongation test, the fabric exhibits three distinct stages: (i) initial elongation: this phase is primarily characterized by internal friction between warp and weft yarns at the intersection points of the fabric structure; (ii) decrimping length: following the initial elongation, the sample's behavior in the central area starts to eliminate the interlacing between warp and weft yarns. Consequently, the warp yarns become independent and taut, extending under the influence of vertical tension; (iii) sample deformation: at this stage, the tensile and elongation load is primarily borne by the decrimped warp yarns, which then represent the vulnerable points of the sample under tension. As a result, the stress–strain curve of the fabric presents an initial segment with lower stiffness, during which the sisal bundles begin to align in the direction of the force. Subsequently, the curve exhibits a linear elastic behavior until the first fibers start rupturing under tension, after which the behavior becomes nonlinear until reaching maximum stress.

It is noteworthy that post rupture of the initial fibers, stress distribution within the fabric is no longer uniform across the cross-section. Instead, stress concentration occurs in the still intact bundles. Figure 5 displays the transverse misalignment of the cotton fiber

following the rupture of the first sisal bundles. This demonstrates that the deformation of the bundles is uneven after the stress peak, with stress relaxation at the edges and increased stress in the central portion of the fabric subjected to tension. An assessment of flax fabrics conducted by Ferrara [20] suggests that the width of the fabric sample under direct tension affects the rupture pattern, which might signify a similar effect.

The deformation, tensile strength, and stiffness values of the reinforcement elements are outlined in Table 4. The fabric demonstrates a tensile strength of approximately 335 MPa, which is lower than the strengths observed in the fiber. Despite this, it remains suitable for utilization in composite materials. In a study by [6], yarn and jute fabric exhibited tensile strengths of 102 MPa and 62 MPa, respectively, in textile-reinforced cementitious composites. Due to its undulating manufacturing process, the fabric exhibits reduced stiffness and increased deformation at rupture compared to the sisal fiber and yarn.

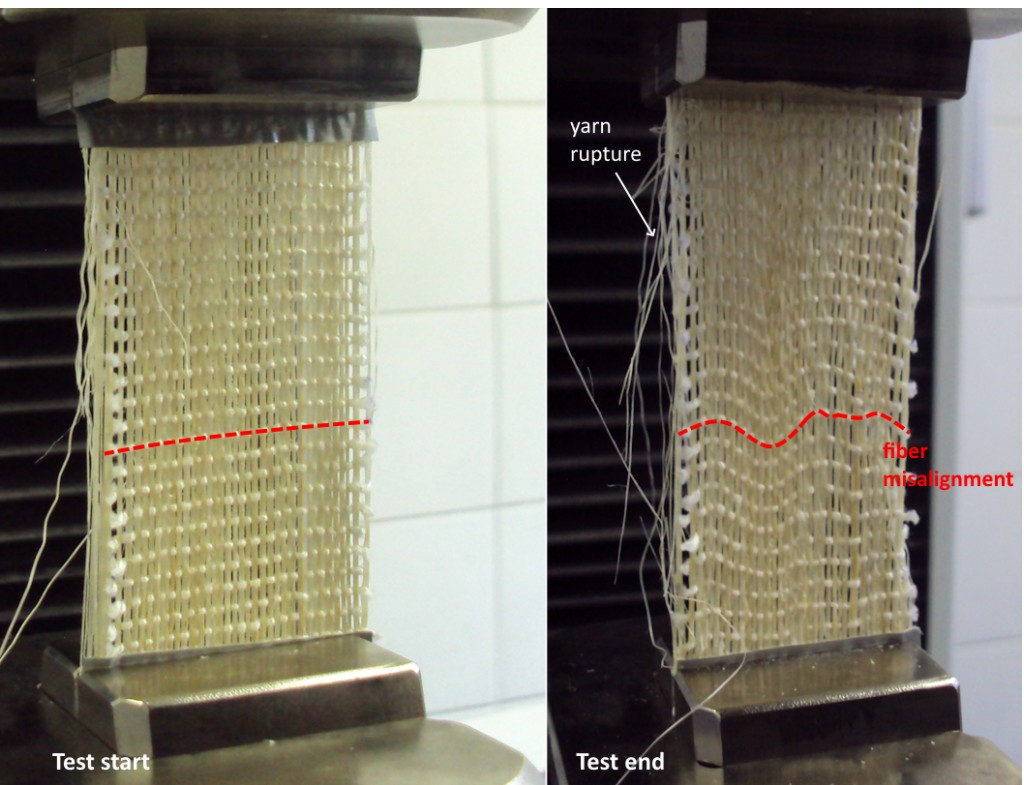

**Figure 5.** Yarn alignment in fabric under tension.

**Table 4.** Tensile test results of the reinforcements.

| Reinforcement | Rupture Strain (mm/mm) | Tensile Strength (MPa) | Stiffness (GPa) |
|---|---|---|---|
| Fiber | 0.0375 ± 0.0060 | 722.11 ± 89.49 | 18.99 ± 1.77 |
| Yarn | 0.0598 ± 0.0073 | 276.50 ± 37.45 | 9.35 ± 1.42 |
| Fabric | 0.1062 ± 0.0118 | 335.77 ± 26.09 | 8.32 ± 0.78 |

*3.2. Tensile Stress-Strain Behaviour of Composite*

3.2.1. Mechanical Behaviour and Failure Mode

Typical examples of stress–strain curves for the sisal fabric-reinforced cement composite (SFRCC) are presented in Figure 6. A strain-hardening behavior is observed for all composites, similar to that observed for composites with vegetal fabric [11], characterized by four stages, as shown in Figure 6: (i) linear-elastic phase; (ii) onset of multiple cracking with maintenance or slight increase in stress; (iii) stress increase and opening of existing cracks; (iv) negative stiffness and loss of strength until failure.

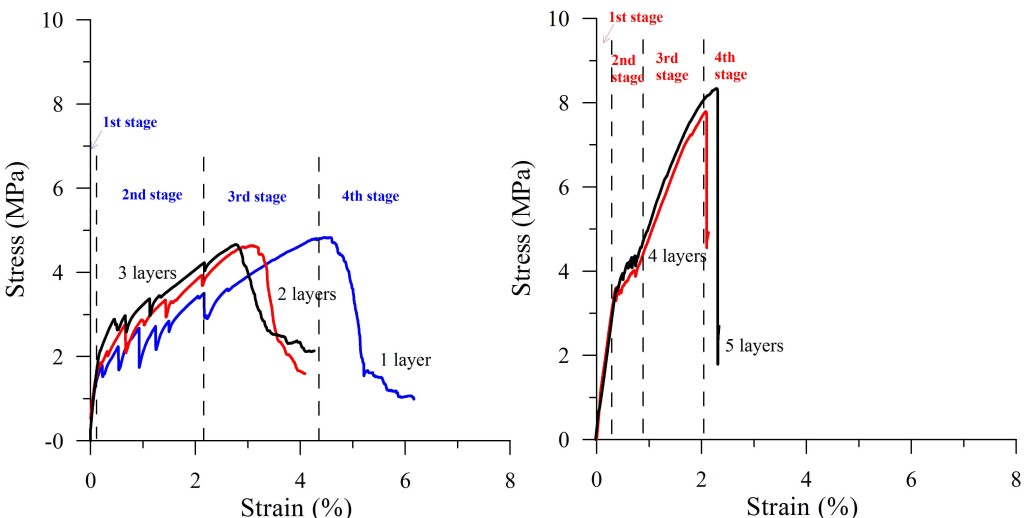

**Figure 6.** Stress–strain curves under tension of representative composite specimens reinforced with one, two, three, four, and five layers of sisal fabrics.

Based on the behavior under direct tension, it is possible to differentiate the composites into two distinct groups: (a) Group I, composites with one, two, and three layers, exhibit a process of multiple cracking with a decrease in stress after the opening of each crack, and a post-cracking stiffness similar to that observed during the multiple cracking process; (b) Group II, composites with four and five layers, showed a significant increase in strength and stiffness after cracking.

In the initial phase (Stage I), the mechanical behavior of the composite is primarily governed by the properties of the matrix. During this period, the stress–strain curve exhibits linearity until the appearance of the first crack. Subsequently, the fibers assume a pivotal role by bridging the surfaces of pre-existing cracks, inhibiting their propagation and mitigating catastrophic failure. This mechanism initiates the formation of multiple cracks within the matrix, dividing it into distinct segments, as depicted in Figure 7. During this phase, the release of stored elastic energy leads to the formation of multiple cracks within the matrix, resulting in an immediate stress drop attributed by [9] to pre-existing cracks in the matrix. A slight stress increase between the initial and final points of this multi-cracking phase, as predicted by [21] and confirmed by various researchers [8,12], is directly linked to the number of reinforcement layers. The culmination of Stage 2 is signified by the absence of new fissures and the consistent spacing between existing cracks. Notably, the spacing between these cracks is directly influenced by the number of layers present within the laminated composites, as shown in Figure 7.

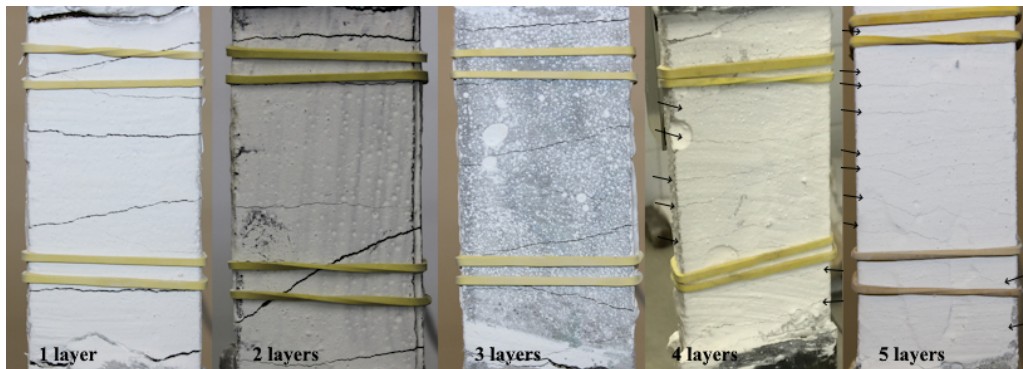

**Figure 7.** Cracking pattern of composites at maximum force (black arrows indicate cracks).

During Stage 3, as the load increases, the fabric experiences stretching, contributing to the curvature observed in the stress–strain curve. As previously discussed, the tensile stress

on the fabric leads to a gradual breakdown of fibers and bundles, resulting in a progressive reduction in the fabric's tensile strength. At the same time, a longitudinal cracking process occurs at the fiber–matrix interface, contributing to the non-linear behavior observed in the stress–strain curve before reaching the maximum stress.

Stage 4 begins when the maximum stress resisted by the composite is reached, initiating the composite's rupture process. Two rupture modes were observed. For Group I composites, an coalescence of the main crack was verified, with gradual tearing of the fabric and extraction from within the matrix due to cracking at the fabric–matrix interface, as shown in Figure 8a,b. For Group II composites, on the other hand, the rupture occurred at higher tensions and abruptly, with the rupture of the covering layer (matrix) that was in contact with the steel plate, as shown in Figure 8c. In detail, a micrograph is presented, in which it is possible to observe that the matrix can infiltrate the spaces between the fibers and between the bundles, ensuring greater adherence and higher composite resistance.

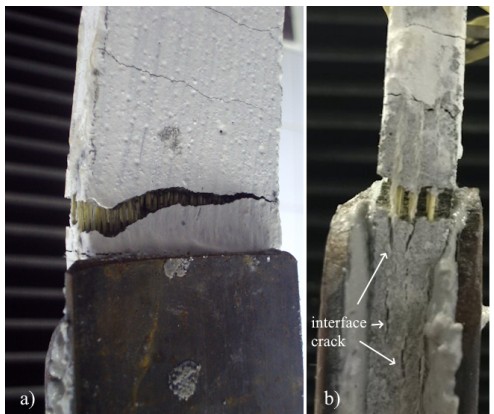 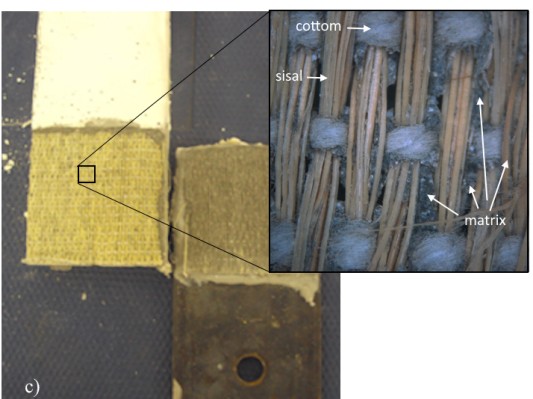

**Figure 8.** Failure mode of the composites under tension: (**a**) coalescence of the main crack; (**b**) fabric pullout and cracking at the fabric–matrix interface; (**c**) rupture of the covering layer.

### 3.2.2. Modelling of Tensile Behaviour of Composites

The determination of the stress–strain diagram of composites under direct tension is crucial for assessing the maximum strength achievable by structural elements. Different approaches to predicting the mechanical behavior of textile-reinforced composites have been proposed in the literature [22]. The ACK model [23], illustrated in Figure 9a, stands out as one of the most commonly employed models [11] due to its ability to linearize the stress–strain behavior, thereby facilitating its implementation. A significant limitation of the model is its assumption of a constant stress during the multiple cracking process, a flaw addressed by Li [12] in the Pseudo Strain-Hardening Model, also depicted in Figure 9a and applied in the modeling of SFRCCs.

Based on the experimental results obtained from three samples for each composite type, the average stress and strain values corresponding to each stage of the stress–strain behavior were determined, as illustrated in Figure 9b.

According to the theoretical model, the value of stress corresponding to the onset of cracking is denoted as $\sigma_{mu}$, indicating the stress at which the matrix reaches the tensile strength within the composite. Experimental results indicate that $\sigma_{mu}$ was approximately 2 MPa (ranging from 1.8 to 2.1 MPa) for Group I composites and around 3.7 MPa (ranging from 3.6 to 3.8 MPa) for Group II composites. This difference suggests that the presence of reinforcement affects matrix cracking.

The effect of the number of layers can also be observed in the multiple cracking process. In Group I samples, this gradual formation of cracks occurred within a strain interval $\Delta_{mc}(=\varepsilon_{mc}-\varepsilon_{mu})$ ranging from 0.018 to 0.021, while for Group II composites, the $\Delta_{mc}$ varied from 0.006 to 0.012. Carreira and Chu [24], in their analysis of the multiple cracking process in concrete beams, found that crack characteristics are influenced by the thickness of the concrete cover over the reinforcement and by the effective tension area of the matrix around

the reinforcement. This phenomenon, known as "tension stiffness," indicates that the stress distribution in the cement matrix is affected by the reinforcement characteristics. Therefore, the values of $\sigma_{mu}$ and $\sigma_{mc}$ cease to be an intrinsic material property and become associated with the distribution of both the matrix and the reinforcement in the structural element.

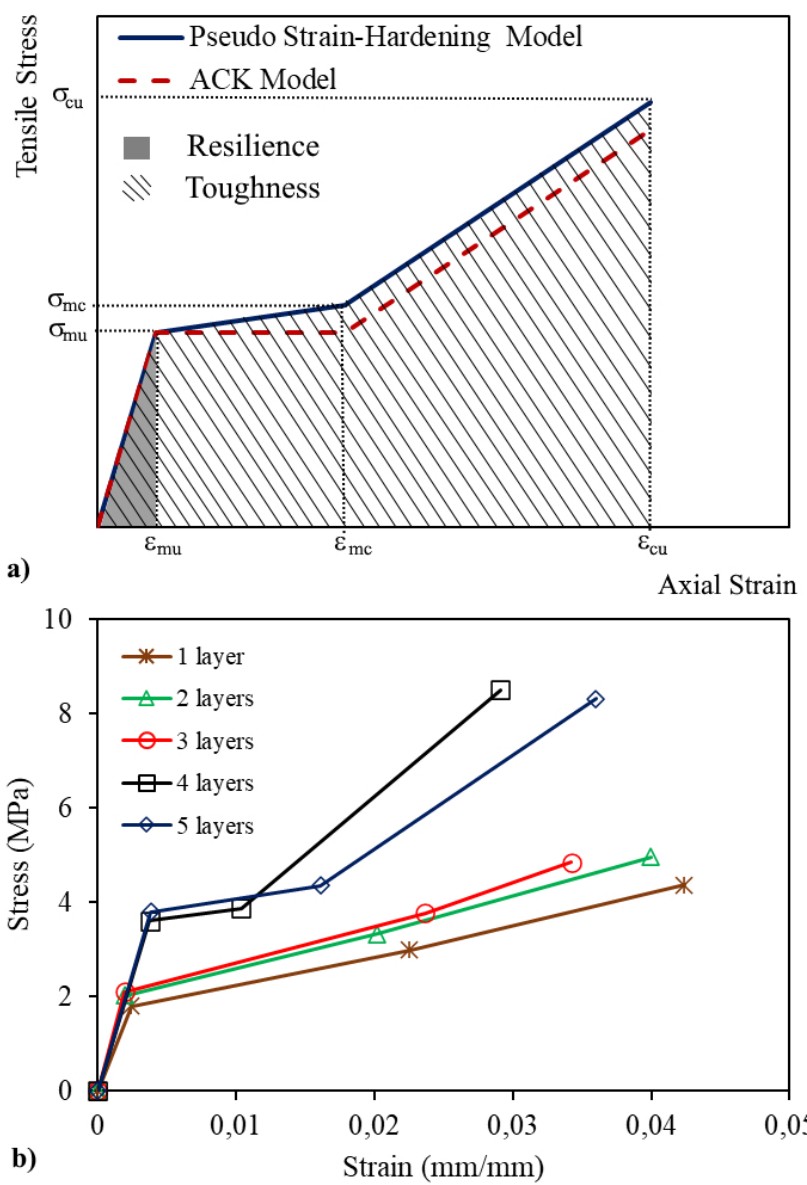

**Figure 9.** Modeling of stress–strain behavior of composites: (**a**) theoretical models; (**b**) linearized experimental results.

The final segment of the diagram reveals that Group II composites, which contain a higher reinforcement content, exhibit greater post-crack stiffness and lower deformation at rupture. In the case of Group I, the final stiffness is roughly similar to that observed during multiple cracking.

### 3.2.3. Tensile Resilience and Toughness

A key aspect of using fabric reinforcement in cement-based materials is the potential to construct more resilient structures. Resilience refers to the material's ability to absorb mechanical energy within its elastic limits, which means its capacity to restore the absorbed mechanical energy and return to its original state. Based on the theoretical stress–strain curve under direct tension, the tensile resilience can be computed by measuring the area

under the curve until the composite reaches the point of its first crack, as depicted in Figure 9a.

Another crucial property of SFRCC is toughness, which denotes its ability to absorb mechanical energy in both elastic and plastic regimes—essentially the total energy the material can absorb while still maintaining functionality. It is represented by the total area under the stress–strain curve. In Figure 9a, the toughness of the composites is displayed, with the total area calculated up to the strain $\varepsilon_{cu}$. Figure 10a showcases the resilience values obtained for the composites.

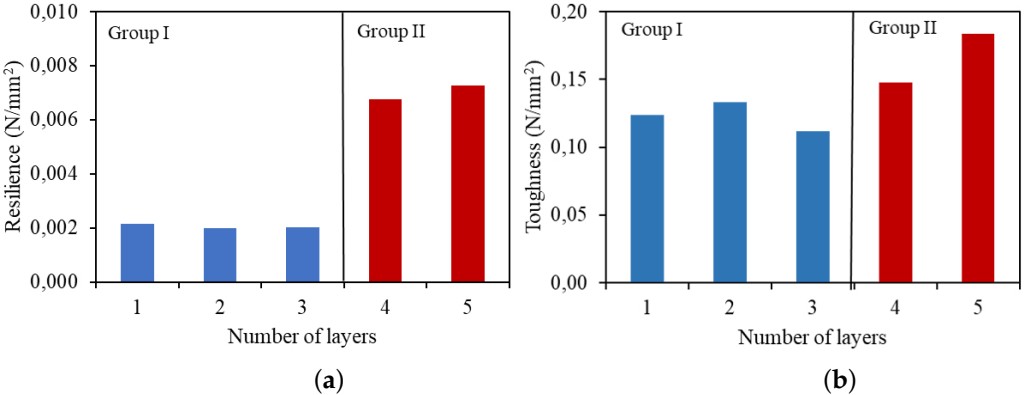

**Figure 10.** Resilience (**a**) and toughness (**b**) of composites under tension.

It is observed that, within the groups, there is no significant variation in resilience. However, there was an increase in resilience of 213% and 236% when the amount of reinforcement increased to four and five layers, respectively, compared to the composite with one layer. The effect of the number of layers on the toughness of the composites is presented in Figure 10b. There is not a direct proportion between the number of layers and toughness. While the Group II composites show higher rupture stress, Group I composites achieve greater deformation before breaking. On the other hand, the composite with five layers exhibits higher toughness, approximately 0.18 N/mm$^2$, which is 48% higher than the toughness of the composites with one layer of reinforcement.

### 3.3. Flexural Behaviour of Composite

3.3.1. Mechanical Behaviour

Most structural elements are subjected to bending stresses, either during their use in buildings or during transportation and assembly stages. Therefore, determining the stress-displacement behavior of the composite, as shown in Figure 11, is crucial to determining its application.

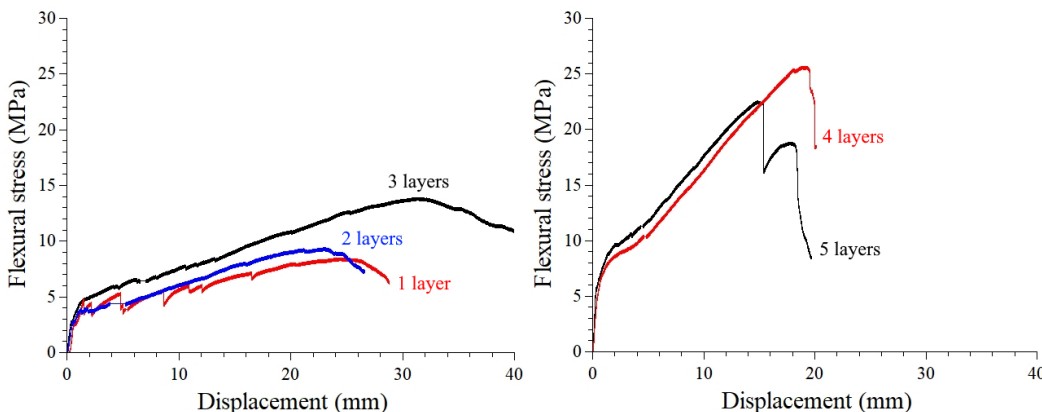

**Figure 11.** Stress-displacement of composites under flexion.

Typical curves obtained from the flexural test for the composites exhibit a behavior like that observed in the direct tension test, with four well-defined stages: a linear elastic region, the appearance of the first crack and multiple cracking (even in composites with one layer), followed by an increase in stress up to the maximum stress point and then a decrease in stress leading to rupture. Similar deflection-hardening behavior under flexion was observed for cementitious laminates reinforced with long sisal fibers [25,26].

The properties of the composites obtained from the bending test are presented in Table 5.

**Table 5.** Flexural test results .

| Sample | First Crack | | End Multiple Crack | | Maximum | |
|---|---|---|---|---|---|---|
| | Stress (MPa) | Deflection (mm) | Stress (MPa) | Deflection (mm) | Stress (MPa) | Deflection (mm) |
| SFRCC1 | 2.4 ± 0.0 | 0.4 ± 0.1 | 4.8 ± 0.9 | 8.3 ± 1.8 | 10.0 ± 0.9 | 24.3 ± 1.2 |
| SFRCC2 | 2.6 ± 0.0 | 0.7 ± 0.1 | 5.9 ± 1.5 | 11.6 ± 2.4 | 8.3 ± 0.2 | 26.5 ± 8.6 |
| SFRCC3 | 2.8 ± 0.3 | 0.4 ± 0.1 | 8.3 ± 0.1 | 25.8 ± 1.2 | 14.4 ± 0.8 | 31.9 ± 0.1 |
| SFRCC4 | 3.9 ± 0.0 | 0.3 ± 0.0 | 10.5 ± 0.5 | 5.9 ± 1.5 | 25.9 ± 0.5 | 17.0 ± 3.5 |
| SFRCC5 | 4.7 ± 0.3 | 0.4 ± 0.0 | 10.8 ± 1.0 | 5.6 ± 0.9 | 18.3 ± 0.1 | 16.1 ± 3.0 |

Concerning the first crack stress, a similar behavior is observed as in the direct tension test results, with little variation among composites within the same group. For Group I, the first crack stress ranged from 2.4 to 2.8 MPa, while higher values were observed for Group II, with 3.9 and 4.7 MPa for the composites with four and five layers, respectively.

Figure 12 illustrates the stress variation of the composites concerning the single-layer reinforced composite. The increase in the number of layers resulted in an increase of up to 95% in the first crack stress. Even though the pre-cracking stage is governed by the properties of the matrix, the presence of the fabric affects the distribution of internal stresses within the composite.

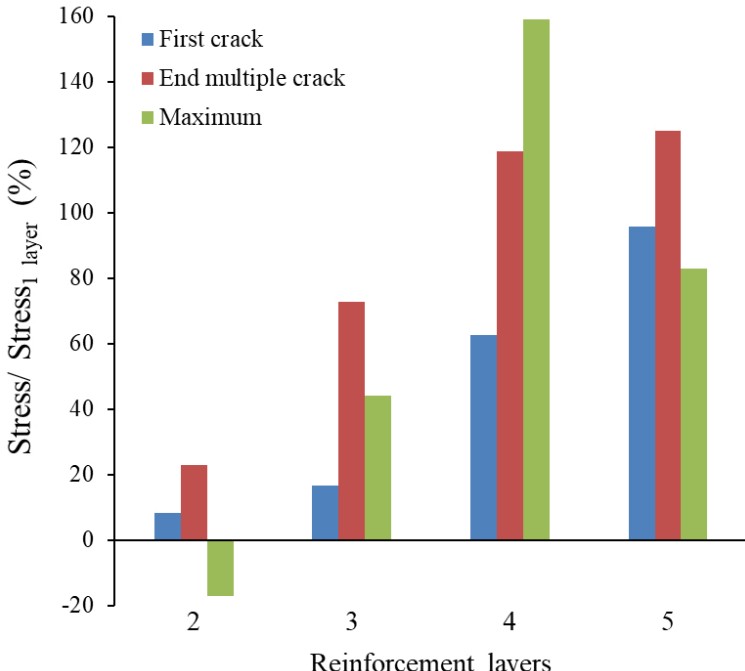

**Figure 12.** Effect of the number of layers on flexural stress relative to the single-layer composite.

Similar to what was observed in direct tension, the multiple cracking process in flexure was accompanied by increased stress for all composites, ranging from 100% to 196% concerning the first crack stress, depending on the number of layers. This increase indicates that the matrix situated between the cracks continues to contribute to the composite's strength. Among the composites, using five layers of reinforcement resulted in higher stress at the end of multiple cracking, with a 125% increase compared to the single-layer reinforced composite.

The maximum stress achieved by the composites was also influenced by the number of layers. However, the increase in the number of layers did not necessarily result in an increase in the strength of the composites, as shown in Figure 12. For composites with two layers, the maximum stress was 17% lower than the stress for the single-layer composite. The composite with five layers showed a reduction of 29% in maximum stress compared to the composite with four layers. This behavior can be explained by stress distribution and the positioning of the neutral stress axis in the cross-section of the composites at the moment of rupture.

Figure 13 presents a schematic drawing with the distribution of deformations and stresses in the reinforcement and matrix at the moment when the compression deformation reaches the peak deformation ($\varepsilon_p$), and the stress on the upper face equals the compressive strength of the matrix (fc), initiating the crushing process. For composites with two or three layers, one of them will be positioned above the neutral axis and, therefore, subjected to compressive stress. Thus, this layer does not contribute to the strength of the composite. In the case of composites with four and five layers, two layers cease to contribute effectively to the final strength of the composite under bending. The placement of symmetrical layers aims to ensure the symmetry of the material, which, in this way, could be subjected to loading on either face. Composites with a greater number of layers, on the other hand, position the lower reinforcement layer closer to the lower face, contributing to a higher stress on the reinforcement.

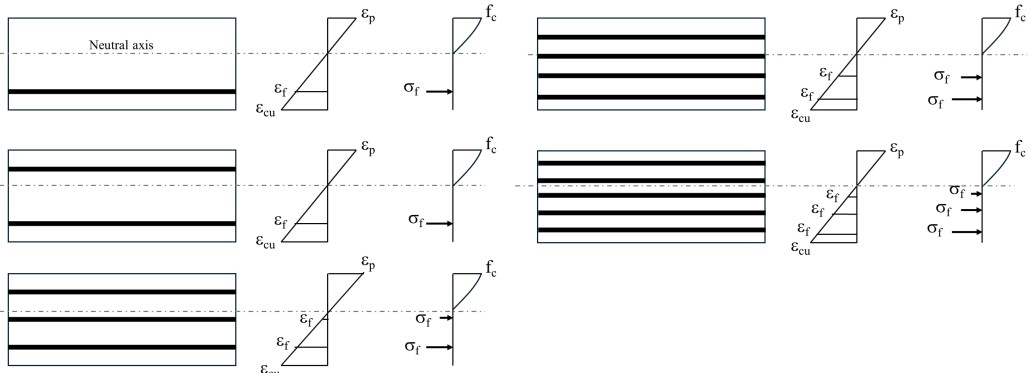

**Figure 13.** Schematic distribution of stress and strain in the cross-sections of composites at the moment of rupture.

## 3.3.2. Cracking and Failure Mode

The process of cracking in the composites under flexural loading was assessed by photographically monitoring the underside of the plates during the loading process. Figure 14 illustrates the cracking on the underside of the Group I composites, while Figure 15 displays the Group II composites at the moment when the maximum flexural stress was reached. It is noticeable that all the composites exhibited multiple cracking and deflection-hardening behavior under flexural stress, with a decreasing variation in crack spacing as the number of reinforcement layers increased. The average crack spacing ranged from 19.0 mm for composites with one layer to 5.9 mm for the composite with five layers.

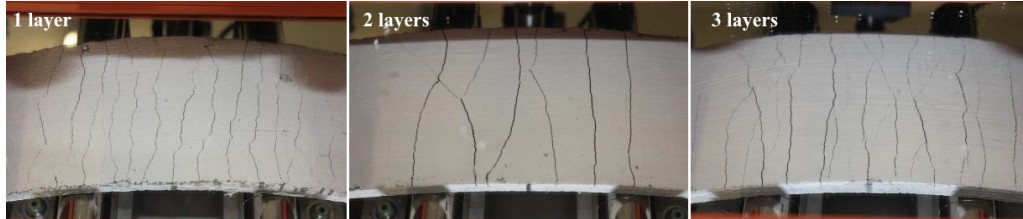

**Figure 14.** Cracking on the bottom face of Group I composites under bending.

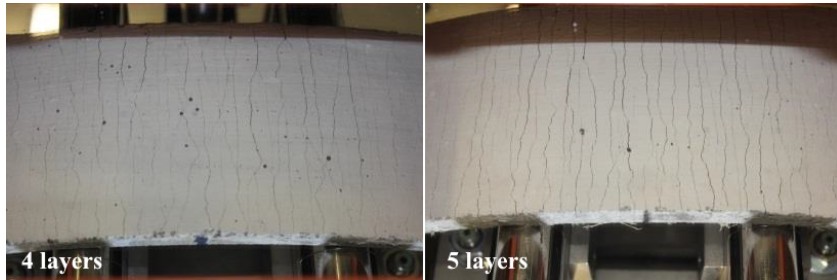

**Figure 15.** Cracking on the bottom face of Group II composites under bending.

The rupture of the composites occurs after significant deformation, as depicted in Figure 16 and Table 5, with displacements at maximum stress ranging from 16 to 32 mm. The maximum value obtained for Group II composites corresponds to a deflection of L/18, which exceeds the maximum allowable value for the serviceability limit state of structural elements, typically around L/150. This deflection limit is based, among other factors, on durability parameters, such as restricting crack widths to prevent the ingress of aggressive agents into the element. However, as evident from Figure 15, even for higher deflections, the cracks exhibit minimal opening due to multiple cracking.

In composites reinforced with polypropylene fabric, Mawlood [7] reported the emergence of longitudinal cracks before flexural rupture, indicating a loss of fabric–matrix adhesion. However, no delamination was observed in the composites produced in this work, indicating that the configuration of the new sisal fabric used was sufficient to allow its impregnation by the matrix, thereby improving fabric–matrix adhesion. As a result, the rupture of the composite occurred through the propagation along the thickness of the plate of the flexural cracks initiated on the lower face, as shown in Figure 16. This led to a translation of the neutral axis (Figure 13) and an increase in deformations and compression stresses, ultimately resulting in the rupture due to the crushing of the matrix on the upper face.

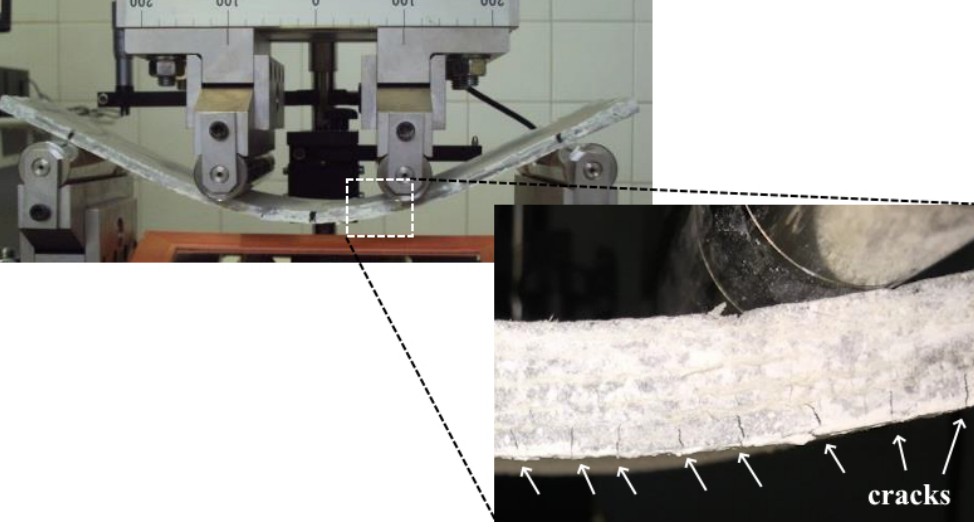

**Figure 16.** Failure mode of composites under bending.

## 4. Conclusions

In this study, a novel sisal fabric composed of straight threads (without twisting) was developed to enhance its suitability as reinforcement for cement-based composites. The research outcomes, derived from evaluating the reinforcement and composites reinforced with one to five layers of fabric, can be summarized as follows:

- Yarns exhibited linear behavior under direct tension, displaying a rapid tension increase post-loading and a ductile rupture due to gradual individual fiber fracture. The fabric also demonstrated ductile rupture, contrary to the individual fiber's brittle rupture, and exhibited sufficient tensile strength suitable for composite reinforcement.
- Composites displayed strain-hardening behavior under direct tension, manifesting multiple cracking and increased stiffness pre-rupture. Greater layering in composites resulted in increased stiffness and post-cracking resistance. Rupture forms varied from crack coalescence and reinforcement pullout in composites with one to three layers, to rupture of the matrix covering layer in composites with four and five layers.
- Stress–strain behavior under direct tension of the composites was modeled using the Pseudo Strain-Hardening Model, enabling determination of resilience and toughness. Composites with five layers demonstrated higher resilience and toughness, although the effect of layering on the latter property was less pronounced.
- Flexural tests on the composites showed deflection-hardening behavior, with increased flexural resistance correlating with the number of reinforcement layers. Composite rupture occurred post multiple cracks and substantial deformation, characterized by crack propagation from the bottom face to the top of the plate.

The values obtained for mechanical strength, resilience and toughness of the composites, particularly when using four and five layers of reinforcement in fabrics, suggest that this more sustainable material can be applied to the production of construction elements, such as tiles, facades, and vertical division elements.

Considering that only one configuration of the new fabric was evaluated in this study, it is still possible to develop new fabrics with different thread thicknesses and spacing between them, so that optimal values can be obtained and more efficient elements can be produced.

**Author Contributions:** Conceptualization, P.R.L.L. and R.D.T.F.; methodology, A.B.d.A.F. and P.R.L.L.; validation, A.B.d.A.F., R.F.C. and O.d.F.M.G.; formal analysis, A.B.d.A.F. and P.R.L.L.; investigation, A.B.d.A.F., R.F.C., P.R.L.L. and O.d.F.M.G.; resources, P.R.L.L., R.F.C. and O.d.F.M.G.; data curation, A.B.d.A.F. and P.R.L.L.; writing—original draft preparation, P.R.L.L. and R.D.T.F.; writing—review and editing, P.R.L.L. and R.D.T.F.; visualization, A.B.d.A.F., R.F.C., P.R.L.L., R.D.T.F. and O.d.F.M.G.; supervision, P.R.L.L. and R.F.C.; project administration, P.R.L.L.; funding acquisition, P.R.L.L. All authors have read and agreed to the published version of the manuscript.

**Funding:** This research was funded by CNPq (MAI DAI Program, Grant number 403644/2020-8) and UEFS (FINAPESQ Program, Grant number 034/2021) for your support.

**Informed Consent Statement:** Not applicable.

**Data Availability Statement:** Data are contained within the article.

**Acknowledgments:** P.R.L.L. thanks CNPq for the postdoctoral fellowship (Grant number 102164/2022-3).

**Conflicts of Interest:** The authors declare no conflicts of interest. The Funders had no role in the design of the study; in the collection, analyses or interpretation of data; in the writing of the manuscript; or in the decision to publish the results.

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
