# Peer review of "Effect of Number of Layers on Tensile and Flexural Behavior of Cementitious Composites Reinforced with a New Sisal Fabric"

_textiles, doi:10.3390/textiles4010004_

Round 1
Reviewer 1 Report
Comments and Suggestions for Authors
This paper experimentally investigate the tensile and flexural behaviors of cement composites with sisal fabric. The effect of number of layers was analyzed and discussed in terms of experimental observation. However, I suggest this paper be rejected due to the reasons as follows.
(1) Only one experimental variable, i.e., number of layer, is considered. Generally, when we presents an experimental study, three or more variables are involved.
(2) Only one specimen is arranged for each test group. At lease three replicated specimens should be set for each group, recommended by many specifications for test.
(3) In the section of "Modelling", I cannot see any models, even one formula.
(4) The conclusions such as linear behavior of yarn, strain hardening behavior, and good mechanical behavior with five layers, are similar with those in other relevant studies. Thus, this study has little contributions for the understanding of cement composites with fabric.
Comments on the Quality of English LanguageMinor editing of English language required
Author Response
This paper experimentally investigate the tensile and flexural behaviors of cement composites with sisal fabric. The effect of number of layers was analyzed and discussed in terms of experimental observation. However, I suggest this paper be rejected due to the reasons as follows.
(1) Only one experimental variable, i.e., number of layer, is considered. Generally, when we presents an experimental study, three or more variables are involved.
Answer. We would like to clarify that it is not just an experimental variable, as there is the development of a new type of fabric with straight threads, which allows for better mechanical behavior than composites produced with twisted threads. In addition, the experimental campaign addresses the characterization of the reinforcement (fiber, yarn, and fabric) and the behavior of 5 composites under direct tension and bending, making the experimental program quite comprehensive.
(2) Only one specimen is arranged for each test group. At lease three replicated specimens should be set for each group, recommended by many specifications for test.
Answer. A standard sampling was used in engineering research with 3 samples per mixture per test, resulting in 30 tested composite samples
(3) In the section of "Modelling", I cannot see any models, even one formula.
Answer. The models are not necessarily presented only through formulas. The equations of the tri-linear model were omitted to avoid extending the article text too much. The equations are presented in the indicated references, but they can be included in the article if the Editor believes that the number of pages can be increased.
(4) The conclusions such as linear behavior of yarn, strain hardening behavior, and good mechanical behavior with five layers, are similar with those in other relevant studies. Thus, this study has little contributions for the understanding of cement composites with fabric.
Answer. No studies were found with 5 layers of plant-based fabric, or fabrics produced with untwisted yarns, in the literature. Furthermore, these are new yarns and fabrics suitable for application as reinforcement in cementitious matrix composites. The objective was to demonstrate their capacity for use as reinforcement in cementitious composite materials. The use of many layers of fabric reinforcement is not common in the literature because existing fabrics are very thick and, unlike what is observed in this article, do not exhibit suitable behavior under tension and flexure.
Reviewer 2 Report
Comments and Suggestions for Authors
This manuscript performs the effect of number of layers on tensile and flexural behaviours of cementitious composites reinforced with a new sisal fabric. The research topic is attractive and practical. However, the following comments should be addressed before a final decision is made:
(1) The novelty of this work must be more explained, which should be highlighted in the introduction part more specifically. In addition, please prove some differences between this paper and the literature.
(2) In the introduction part, the description of related research work mainly covers fabric layers and the influence on different mechanical behaviors. The author should also focus on some finite element models or analytical models that can describe the negative stiffness. It is recommended that the authors add some research work related to this topic.
(3) The quality of some figures in this manuscript should be carefully improved. Some necessary analysis needs to be conducted.
(4) The contents described in the conclusion section only states some findings in the experiment. The related limitations should be summarized.
(5) The authors are invited to make a comment on how to make good use of fabric in reinforcing cement-based materials.
(6) The manuscript includes some typos, punctuation, and grammatical problems. For example, the reviewer is very confused about tense problems in this manuscript. The whole paper should be rechecked carefully.
Comments on the Quality of English LanguageMinor editing of English language required
Author Response
- The novelty of this work must be more explained, which should be highlighted in the introduction part more specifically. In addition, please prove some differences between this paper and the literature.
Action: In the introduction and discussion of the results, studies already developed and the differences between them and our article were presented.
- In the introduction part, the description of related research work mainly covers fabric layers and the influence on different mechanical behaviors. The author should also focus on some finite element models or analytical models that can describe the negative stiffness. It is recommended that the authors add some research work related to this topic.
Answer. As the focus of the article is not on the modeling of composites, a detailed study of the different models that can be used was not pursued.
Action: The introduction has been improved. A brief description of the mechanical behavior with indications of methods for evaluating negative stiffness was inserted in the introduction.
(3) The quality of some figures in this manuscript should be carefully improved. Some necessary analysis needs to be conducted.
Action: The figures have been reviewed, and some have been improved, as well as certain discussions
(4) The contents described in the conclusion section only states some findings in the experiment. The related limitations should be summarized.
Action. The conclusion has been modified.
(5) The authors are invited to make a comment on how to make good use of fabric in reinforcing cement-based materials.
Action. The conclusion has been modified.
(6) The manuscript includes some typos, punctuation, and grammatical problems. For example, the reviewer is very confused about tense problems in this manuscript. The whole paper should be rechecked carefully.
Action. A new review of the text has been carefully conducted to address the indicated errors
Reviewer 3 Report
Comments and Suggestions for Authors
The authors have produced a very interesting and highly applicable work for the construction sector. Only a few minor comments are included for possible correction before publication in the journal.
If possible, use units in the international system. Also, be careful with superscripts (g/cm3, and others).
What was the standard of mixing and consistency used to make the mortar?
How is slippage between the fibres prevented when flexural breakage occurs?
Figure 10 and 12, include the error.
Figure 13 is very presumptuous, could the authors guarantee this deformation behaviour, if not include the limitations of the model.
Improve the discussion of Figure 16 and address the implications of the study.
Include limitations and future lines of work in the conclusions.
Author Response
(1) If possible, use units in the international system. Also, be careful with superscripts (g/cm3, and others).
Action. Corrected
(2) What was the standard of mixing and consistency used to make the mortar?
Action. The information has been inserted into the methodology.
(3) How is slippage between the fibres prevented when flexural breakage occurs?
Answer: As indicated in the text, no cracking of the fiber-matrix interface was observed in the final stages of the bending test, as shown in Figure 16. It is believed that the use of meshes with proper spacing between threads allowed the matrix to penetrate the fabric and provide mechanical anchorage.
Action: The text has been improved to make the interaction between fabric and matrix clearer.
- Figure 10 and 12, include the error.
Answer: The values presented in Figure 10 were obtained from the theoretical stress-strain curve, so the resilience and toughness values are theoretical values. Figure 12 is a relationship between the stress in composites with 2 to 5 layers relative to the composite with 1 layer. It is a theoretical value obtained with the average values.
(5) Figure 13 is very presumptuous, could the authors guarantee this deformation behaviour, if not include the limitations of the model.
Answer: The figure was defined assuming a balanced rupture of the plate matrix, which occurs when the matrix is crushed in the compressed region (i.e., when the compression deformation reaches the peak deformation) and cracking in the tensile region of the matrix. Thus, the position of the neutral axis was calculated.
Action: The text was modified to clarify that it is a schematic drawing.
(6) Improve the discussion of Figure 16 and address the implications of the study.
Action: Corrected
(7) Include limitations and future lines of work in the conclusions.
Action. The conclusion has been modified.
Round 2
Reviewer 1 Report
Comments and Suggestions for Authors
The paper has been revised according to reviewer's comments
Reviewer 2 Report
Comments and Suggestions for Authors
The entire paper is clearly presented and suitable for publication in its current form.
Reviewer 3 Report
Comments and Suggestions for Authors
The authors have made significant changes to the manuscript and have taken the reviewer's comments into consideration. The article deserves to be accepted for publication.
The authors are recommended to read the following paper, which may be useful for future research:
https://doi.org/10.1016/j.compositesb.2019.107390